# Insights on pathoadaptation of sequential *Pseudomonas aeruginosa* isolates to the urinary tract

Caroline Martin-Duval,[1] Sandrine Dahyot,[2] Inès Coquisart,[3] Benoit Bernay,[4] Martine Pestel-Caron,[2] Jean-Christophe Giard[3]

**ABSTRACT** *Pseudomonas aeruginosa* is an opportunistic pathogen responsible for 10% of nosocomial urinary tract infections (UTIs). Its large genome and adaptability enable it to cause a wide variety of infections, from respiratory disorders in cystic fibrosis patients to recurrent UTIs. While genomic and phenotypic adaptations of *P. aeruginosa* have been well-studied in respiratory infections, few studies have investigated pathoadaptation in recurrent UTIs. Here, we investigated the impact of genomic alterations of sequential urinary isolates collected from three patients on phenotypic responses to environmental stresses, virulence, and motility. In addition, to gain insight on adaptive mechanisms of *P. aeruginosa* in urine, proteomic analyses were conducted using pooled human urine compared to a standard Trypticase Soy (TS) medium. Late isolates showed significantly impaired growth and reduced responses to acid and osmotic stresses compared to early isolates, although responses to oxidative stress remained unchanged. Furthermore, the late isolates were significantly less virulent in the *Galleria mellonella* infection model. Finally, proteomic analyses revealed the accumulation of proteins associated with flagellum and chemotaxis only in early isolates of two out of three patients, regardless of the culture medium. Motility assays confirmed these results, with late isolates being less motile than early ones. Moreover, siderophore-related proteins were significantly less abundant in late isolates when cultured in human urine, a result not observed in TS medium, suggesting a convergent adaptation trajectory in urine. Our results provide an initial insight into the adaptive mechanisms of *P. aeruginosa* in the urinary tract.

**IMPORTANCE** *P. aeruginosa* is the third most common pathogen causing healthcare-associated UTIs. Its ability to form biofilms and develop antibiotic resistance often leads to relapses. Here, we investigated the phenotypic characteristics of longitudinal urinary isolates under specific stress conditions and used an *in vivo* model to evaluate the virulence of these isolates. Integrating proteomic analysis into this approach allowed us to identify the metabolic and regulatory pathways involved in bacterial adaptation and to establish innovative correlations between genomic, phenotypic, and proteomic data. Altogether, these data enabled us to map the adaptation mechanisms of *P. aeruginosa* in the urinary environment. These findings provide putative new therapeutic targets and contribute to our understanding of recurrent UTIs caused by this pathogen.

**KEYWORDS** *Pseudomonas aeruginosa*, urinary tract infection, adaptation, stress response, virulence, proteomics

*P*seudomonas aeruginosa is an opportunistic Gram-negative rod-shaped bacterium that can cause a wide range of infections with high morbidity and mortality rates (1). Due to its metabolic versatility and numerous virulence factors (2), *P. aeruginosa* is responsible for life-threatening acute and chronic infections, such as cystic fibrosis (CF) pulmonary disease, burn wound infections, and urinary tract infections (UTIs) (1, 3). UTIs are among the main manifestations of nosocomial infections, with *P. aeruginosa*

**Peer Reviewer** Sébastien Bontemps-Gallo, Centre d'Infection et d'Immunite de Lille, Lille, France

Address correspondence to Caroline Martin-Duval, caro.27950@gmail.com.

Martine Pestel-Caron and Jean-Christophe Giard contributed equally to this article.

The authors declare no conflict of interest.

responsible for approximately 9% of cases in acute care hospitals, according to the latest point prevalence survey of healthcare-associated infections and antimicrobial use in European acute care hospitals (4). The ability of this species to form biofilm and its multidrug resistance are key factors explaining the treatment difficulties and the frequent relapses and chronicity of *P. aeruginosa* infections (5).

The adaptation mechanisms of *P. aeruginosa* isolates during chronic infections have been studied for many years, notably in the context of long-term respiratory tract infections in CF patients. Large genomic deletions affecting up to 8% of the genome have been reported (6), particularly in virulence-associated loci (7). However, studies on the adaptation mechanisms of *P. aeruginosa* in the urinary tract remain scarce (8, 9). A previous work of our team on sequential urinary isolates from recurrent infection or colonization in seven patients revealed within-host evolution through large genomic deletions (ranging from 32 to 365 kb) and single nucleotide polymorphisms (SNPs) ranging from 0 to 126 SNPs per genome per year between early and late isolates collected from a given patient (10). These genomic adaptations occurred preferentially in genes involved in carbon compound catabolism, transcriptional regulation, and two-component systems. In addition, phenotypic changes with reduced fitness of late isolates in trypticase soy (TS) and artificial urine medium (AUM) have been characterized (10). The objective of this study is to further characterize *P. aeruginosa* mechanisms of survival and adaptation in the urinary tract by comparing the behaviors of three pairs of urinary isolates (one early and one late with large genomic deletions) from three patients. For this purpose, we assessed the impact of genomic changes on growth in diverse media (*in vitro* rich medium, AUM, and pooled human urine [HU]), responses to various stresses (osmotic, acidic, and oxidative stresses), and virulence (using the *Galleria mellonella* infection model). In parallel, a proteomic approach was carried out to identify the metabolic pathways involved in *P. aeruginosa* adaptation to the urinary environment over time.

## RESULTS

### Isolates selection

This study included a set of six isolates from three patients (A, D, and F, whose late strains presented large deletions) (Fig. 1). For each patient, we sought to compare the phenotypic characteristics of one early isolate (A-e, D-e, and F-e) obtained from the first urine sample collected with that of a late isolate (A-l, D-l, and F-l) obtained from the last urine sample, in order to describe intra-patient bacterial adaptation and correlate them with the genomic changes previously identified (10).

### Growth assays

Growth curves of early and late isolates were compared under three conditions: rich medium (TS), HU, and AUM. Final counts and $OD_{600}$ of early and late isolates from the three patients after 24 h of culture in TS were reported in Table S1. The late isolates from patients D and F exhibited significantly reduced growth in all three media compared to their early counterparts (Fig. 2B and C): the generation times of isolates D-l and D-e were 48 ± 6 min vs. 25 ± 4 min in TS, 65 ± 17 min vs. 23 ± 5 min in HU, and 71 ± 3 min vs. 34 ± 2 min in AUM, respectively (Table S2). The difference in growth rates between the early and late isolates from patient D was less pronounced in HU than in TS, with growth reaching stationary phase after 270 ± 30 min vs. 420 ± 54 min in TS and 235 ± 9 min vs. 360 ± 30 min in HU (Table S2). In contrast, the late isolates from patient A (A-l) grew significantly faster in HU (40 ± 1 min) and AUM (31 ± 4 min) compared to the early isolates (53 ± 2 min in HU and 43 ± 2 min in AUM) ($P < 0.01$ and $P < 0.05$, respectively) (Fig. 2A). Of note, the early and late isolates from patient A showed no significant difference in growth rate in TS (33 ± 5 min for the early isolate and 33 ± 3 min for the late isolate). Moreover, comparison of the bacterial growth in HU and AUM showed significantly ($P < 0.05$) different generation times for three isolates (A-e, D-e, and F-l). These

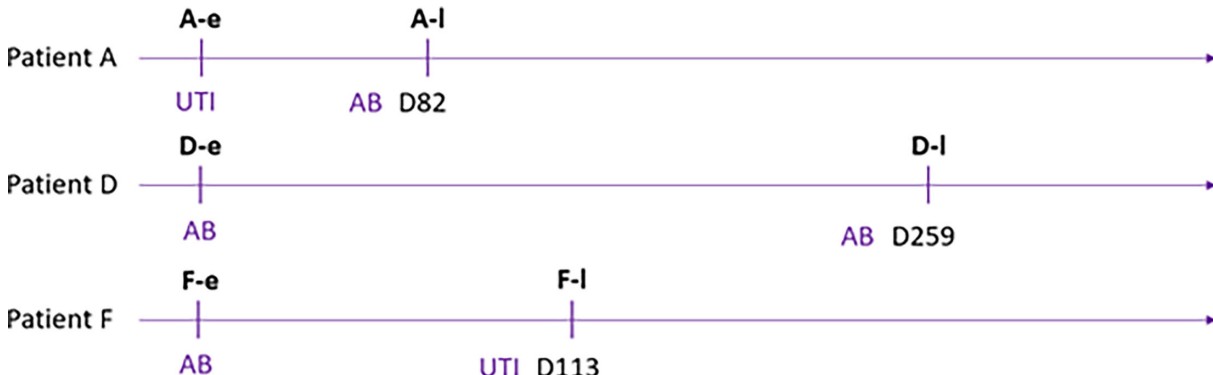

**FIG 1** Flowchart of the six *P. aeruginosa* urinary clinical isolates of this study. Each isolate was associated with a context of asymptomatic bacteriuria (AB) or urinary tract infection (UTI) (10). The number of days (D) between the first and the second sample is indicated for each patient. e, early isolate; l, late isolate.

data pointed out that the ability of *P. aeruginosa* to grow in urine may decrease or increase after prolonged colonization in the urinary tract, up to 259 days, depending on the patient.

## Stress responses

To investigate the impact of large genomic deletions in late isolates on environmental stress responses, growth of isolates was monitored under osmotic, acidic, and oxidative stress conditions.

Late isolates were more susceptible to osmotic stress compared to the early ones, particularly in the presence of 0.3 and 0.4 M NaCl, for patients D and F (Fig. 3; Table S3). In the case of patient D, the generation time of D-l increased from 48 ± 6 min in the control medium to 67 ± 5 min with 0.4 M NaCl ($P = 0.005$), while no significant difference was observed for the early isolate in the presence or absence of NaCl (Fig. 3; Table S3). For patient F, the growth rate of F-l increased by 70% between the condition without NaCl and 0.4 M NaCl, compared to a 37% increase for the early isolate (F-e) ($P = 0.0023$) (Fig. 3C; Table S3). Conversely, for patient A, A-l generation time increased significantly only at 0.2 M NaCl compared to the control medium (47 ± 4 min vs 41 ± 8 min) ($P = 0.0101$). The lower impact of osmotic stress on isolates from patient A may be associated with the lower growth capacity of the early isolate in the control medium compared to those from patients D and F (33 ± 5 min vs 25 ± 4 min and 27 ± 1 min, respectively). These findings demonstrated that osmotic pressure had variable impact on the growth of *P. aeruginosa* urinary isolates depending on the patient and the NaCl concentration.

As observed in Fig. 4, a significant increase in generation time at pH decreased from 7 to 5 was significantly more pronounced for the late isolates from patients A and F (34% and 36% increases, respectively) ($P = 0.03$ and $P = 0.028$, respectively) than for the late isolate from patient D (18% increase) ($P = 0.03$) (Table S4).

Our data demonstrated that the reduction of growth was more pronounced in late isolates, although early isolates were sensitive to osmotic stress and pH variations.

Interestingly, no significant changes in growth rate were observed for any of the isolates under oxidative stress conditions (hydrogen peroxide concentrations ranging from 0.1 to 0.4 mM $H_2O_2$) (Fig. 5; Table S5). Taken together, these results showed that the impact of stresses was strain-dependent and mainly led to a slower growth.

## Virulence characterization

To compare the virulence of early and late isolate pairs of *P. aeruginosa*, a *G. mellonella* model of infection was used.

The early isolates from patients A and D demonstrated a significantly greater capacity to kill larvae compared to their corresponding late counterparts ($P = 0.014$ and $P = 0.01$, respectively) (Fig. 6; Table S6). While the virulence of the late isolate from patient F also

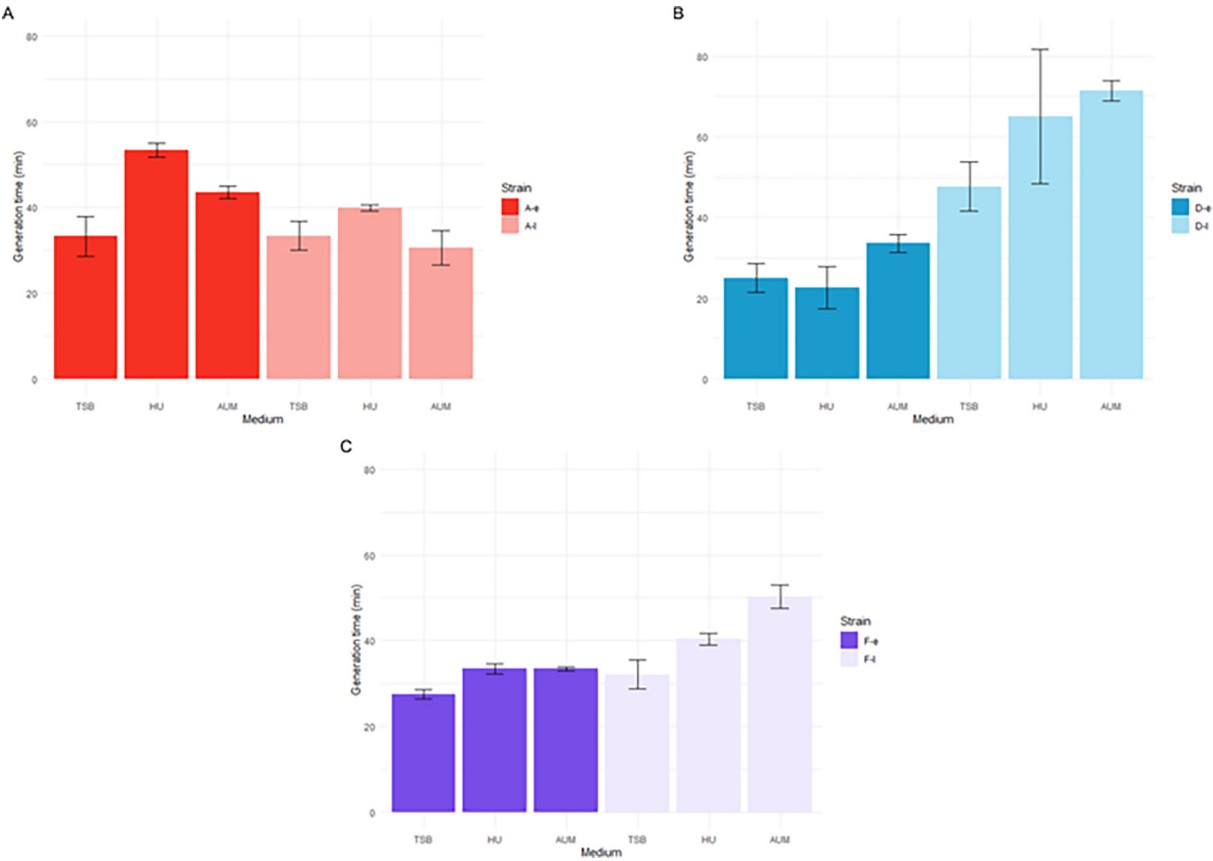

**FIG 2** Generation time of early and late isolates from patients A (A), D (B), and F (C) grown in trypticase soy broth (TS), human urine (HU), and artificial urine medium (AUM). Error bars represent the standard deviations of at least three independent experiments. Statistical significance was determined as *P* < 0.05 using a two-sample Student's *t*-test for independent samples.

decreased, this reduction was not statistically significant. Between 90% to 100% of the larvae were still alive after 48 h after infection with the late isolates of the three patients.

## Proteomic profiles

The proteomes of the three pairs of early and late isolates grown in TS, AUM, and HU media were analyzed by mass spectrometry to compare differentially expressed proteins and the metabolic pathways according to the medium, and to identify markers of adaptation. Comparison of proteomic profiles of bacterial cells grown in HU and AUM enabled us to confirm that AUM did not correctly mimic HU, as strong dissimilarities were observed in the proteins accumulated in these two media (Table S7, Fig. S1). Thus, we focused exclusively on the comparison between TS and HU for the rest of the study.

Among the 3,276 total proteins identified across the six isolates and the two media, 1,602 polypeptides exhibited differential abundances. It should be noted that proteins corresponding to genes deleted in the late isolates (10) were not included in subsequent analyses. Proteins were grouped based on their functional categories using the Kyoto Encyclopedia of Genes and Genomes (KEGG) database.

The number of functional categories shared or unique to early and late isolates from the three patients, in TS and in HU, is illustrated in Fig. 7. Venn diagrams showed that the diversity of metabolic categories impacted between early and late isolates was patient-dependent. In HU, overabundant proteins identified in A-l, D-l, and F-l belonged to 11, 25, and 42 metabolic pathways, respectively (Fig. 7). Early and late isolates from patient A displayed a lower number of common metabolic pathways than those of patients D and F in HU. Thus, they shared few categories (*n* = 6) compared to patient F whose isolates

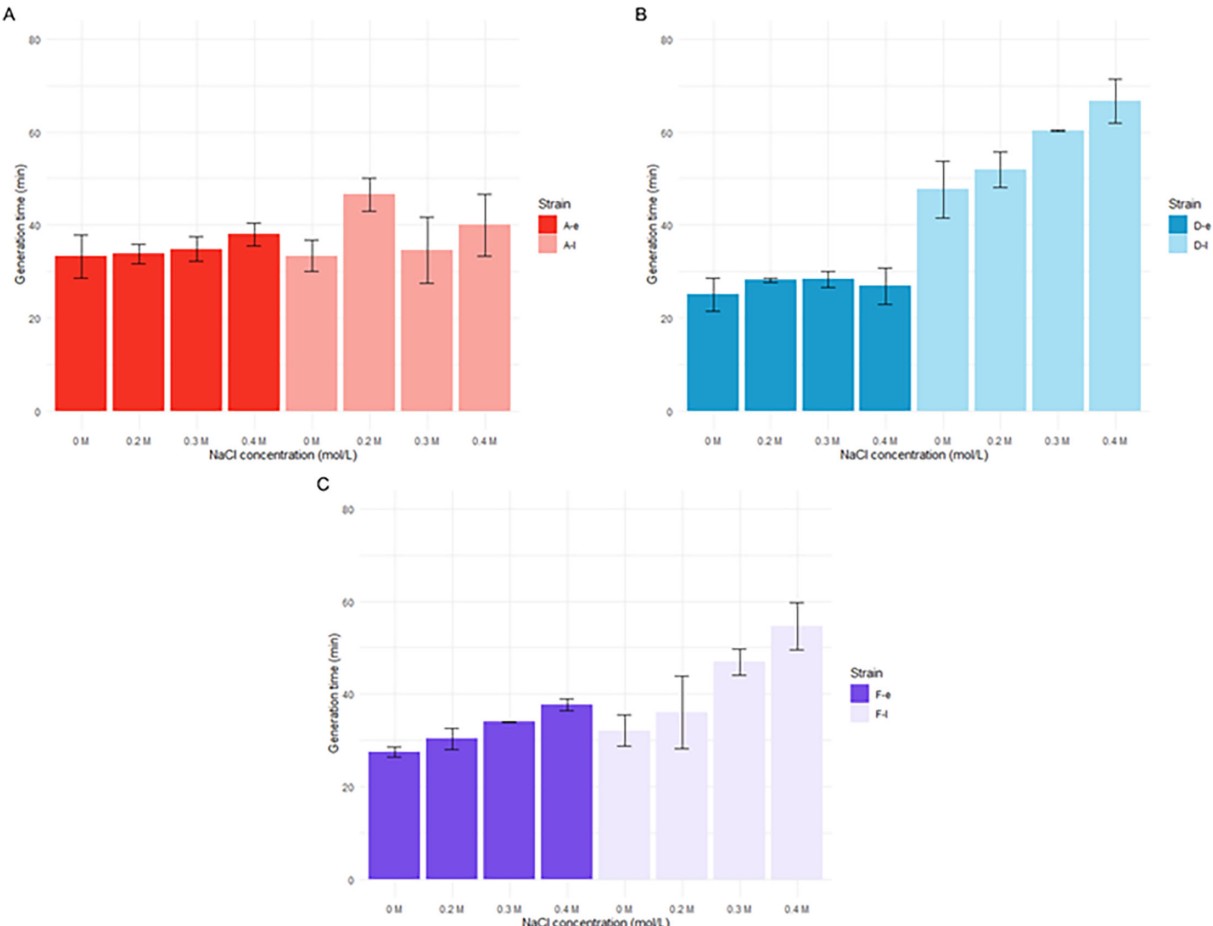

**FIG 3** Generation time of early and late isolates of patients A (A), D (B), and F (C) grown in trypticase soy broth (TS) with different NaCl concentrations (0, 0.3, and 0.4 M). Error bars represent the standard deviations of at least three independent experiments. Statistical significance was determined as $P < 0.05$ using a two-sample Student's $t$-test for independent samples.

shared the most of the categories identified ($n = 30$). Proteomic data showed that few proteins were conserved across patients, but it was mainly in terms of metabolic categories that some of them were found in common. Finally, there were three shared metabolic pathways (biosynthesis of siderophores, ribosome, and oxidative phosphorylation) between early isolates from patients A, D, and F in HU, while any common pathway was found in late isolates (Fig. S2).

The overabundant proteins in the late isolate (A-l) of patient A grown in HU belonged to only 11 distinct metabolic pathways (Fig. 8A), primarily linked to energy metabolism, such as glycolysis/gluconeogenesis and glyoxylate metabolism. Moreover, proteins belonging to biofilm formation and bacterial secretion system categories, which are considered virulence factors, were the most overabundant in the early isolate A-e. For patient D (Fig. 8B), the overabundant proteins accumulated by the late isolate D-l were involved in more various metabolic pathways. As for the late isolate of patient A, overabundant proteins were associated with energy metabolism, such as glycolysis, tricarboxylic acid cycle, and glyoxylate metabolism. The early isolate D-e exhibited overabundant proteins involved in chemotaxis, biofilm formation, and amino acid metabolism. An overabundance of proteins involved in several energy metabolisms was found in both early and late isolates from patient F. However, enzymes associated with amino acid metabolisms were mainly identified in the early isolate F-e, while proteins linked to biofilm formation were identified in the late isolate F-l.

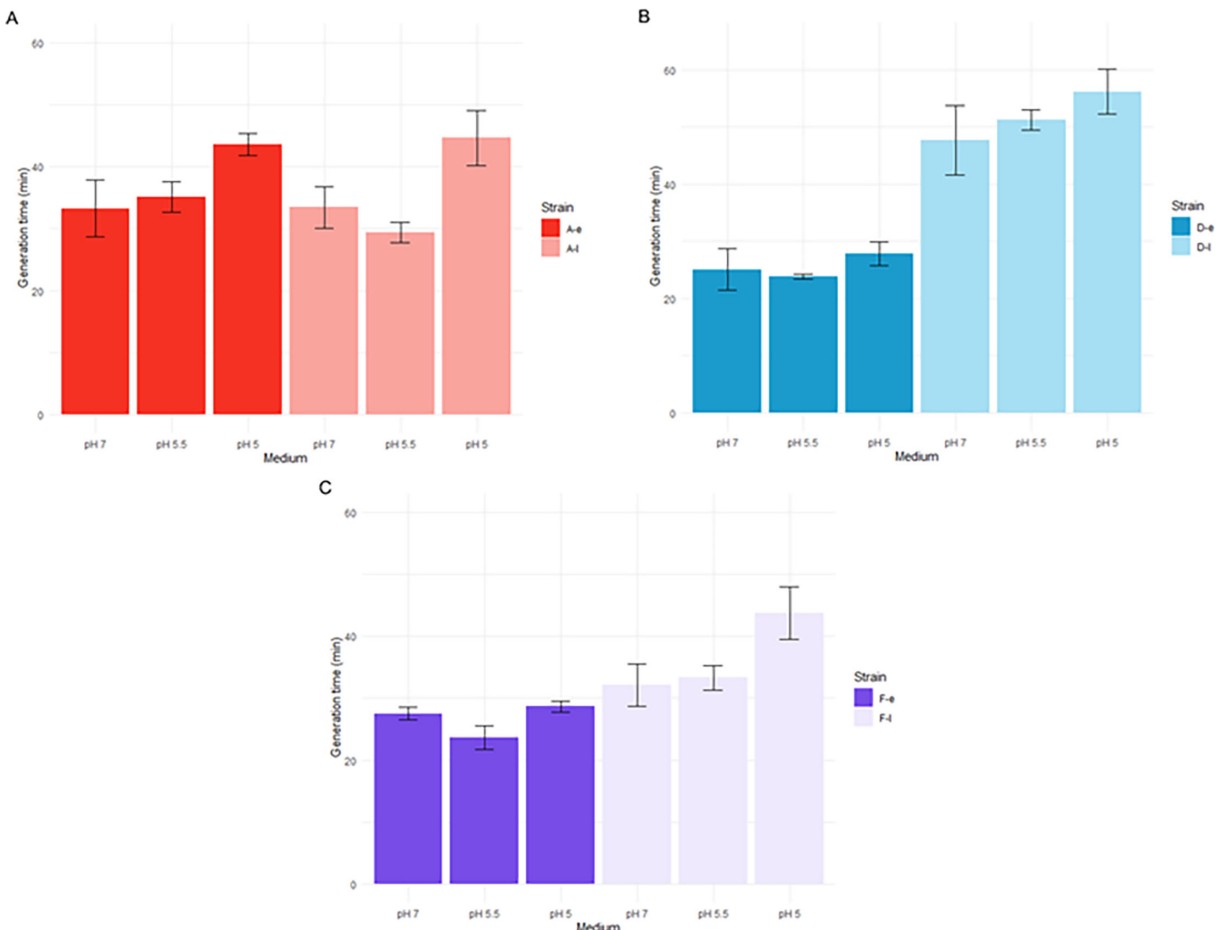

FIG 4 Generation time (min) of early and late isolates from patients A (A), D (B), and F (C) grown in trypticase soy broth (TS) at pH 7, 5.5, and 5. Error bars represent the standard deviations from at least three independent experiments. Statistical significance was determined as *P* < 0.05 using a two-sample Student's *t*-test for independent samples.

ABC transporters and two-component systems (TCS) were the two categories for which the number of proteins differentially expressed was the highest in the early isolates of patients D and F. Of note, the OpuCA ABC transporter protein, involved in osmoprotectant transport and enhancing the cellular hyperosmotic salinity response, was overabundant in the late isolate D-l grown in HU (Table S7). Similarly, the OpuCD ABC transporter and a protein involved in glycine-betaine transport were overabundant in the late HU-grown isolate F-l.

Surprisingly, no proteins involved in the biosynthesis of pyoverdine and pyochelin, the two main siderophores of *P. aeruginosa* (11), were overabundant in the HU-grown late isolates of patients A, D, and F, whereas 13 to 14 such proteins were overabundant in their early counterparts (Table 1).

Similarly, proteins involved in siderophore-independent iron acquisition, such as heme oxygenase (HemO) and the Fe(III) dicitrate transport protein (FecA), were exclusively overabundant in early isolates grown in urine (Table S7). Interestingly, an overabundance of proteins involved in flagellar assembly as well as proteins related to chemotaxis was mainly observed in early isolates grown in both TS and HU media (Table 1).

## Motility assays

To test whether late isolates were less motile than early ones, as suggested by the proteomic profiles, motility assays on TS agar plates were performed using the *P.*

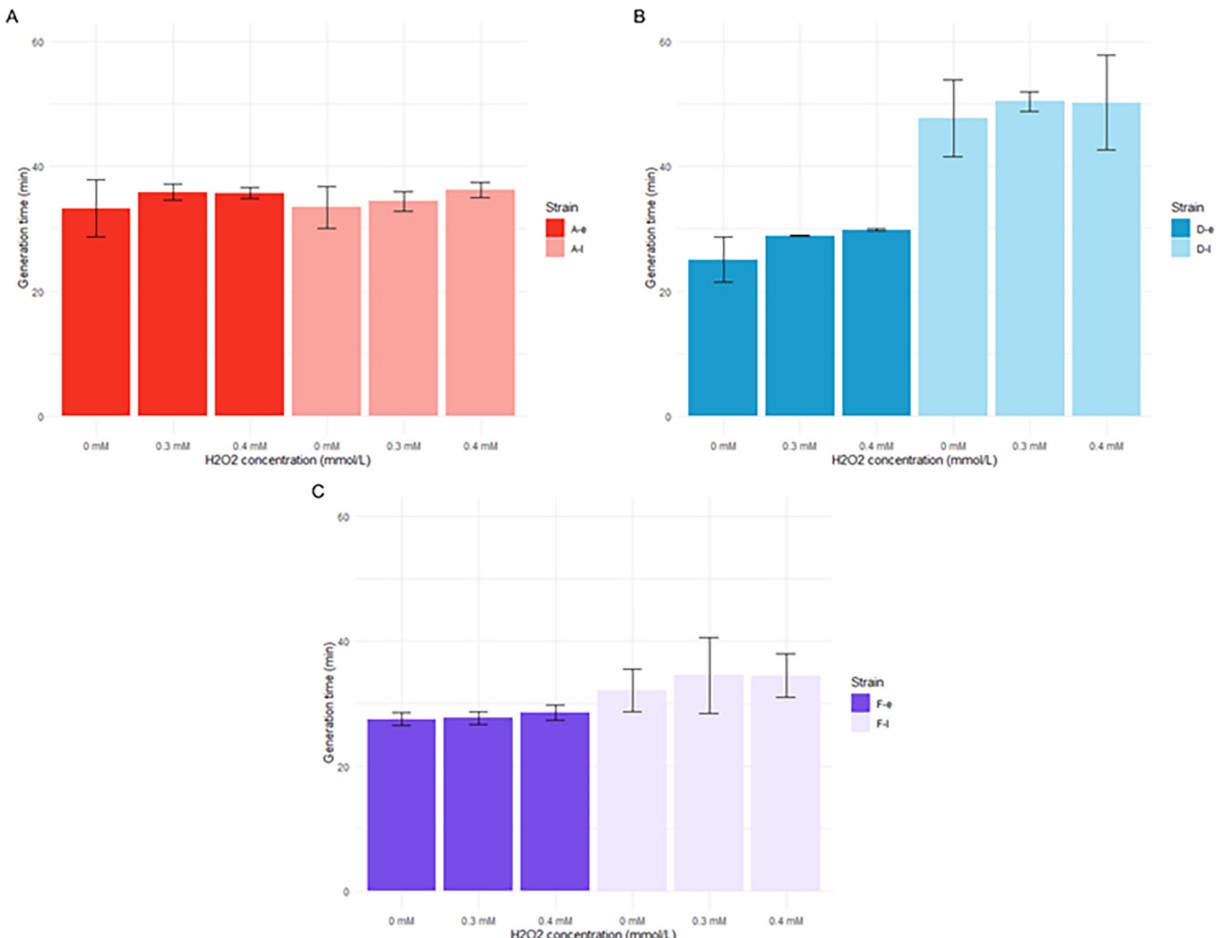

**FIG 5** Generation time of early and late isolates from patients A (A), D (B), and F (C) grown in trypticase soy broth (TS) under different $H_2O_2$ concentrations (0 mM, 0.3 mM. and 0.4 mM). Error bars represent the standard deviations from at least three independent experiments. Statistical significance was determined as $P < 0.05$ using a two-sample Student's $t$-test for independent samples.

*aeruginosa* reference strain PA14 as a positive control, since it displays both swimming and swarming motilities (Fig. S3). All isolates from the three patients, except D-e, showed significantly reduced swimming compared to PA14 ($P < 0.001$) (Fig. 9). Moreover, late isolates from patients D and F completely lost this motility compared to early isolates, whereas the swimming ability of A-e and A-l remained the same (10 ± 2 mm *vs* 10 ± 1 mm). These results were consistent with proteomic data where flagellum- and chemotaxis-associated proteins were much more abundant in early isolates of patient D and F, but not in patient A (Table 1). Swarming motility, which also requires a functional flagellum, was tested in semi-solid medium Fig. S3. All isolates from the three patients displayed a swarming capacity comparable to PA14, except for the late isolate from patient D, which completely lost this motility, likely due to deletions of the genes encoding the proteins involved in flagellum/chemotaxis (Fig. 9).

Finally, a summary table with the major phenotypic characteristics of late isolates was presented (Table 2).

## DISCUSSION

*Pseudomonas aeruginosa* causes UTIs in patients with urinary tract abnormalities or urinary catheters (2, 12). Such UTIs are often associated with recurrence, antibiotic resistance, and longer hospitalization (1), emphasizing the importance of enhancing our knowledge of its adaptive evolution. To date, no study has specifically analyzed the growth of *P. aeruginosa* in human urine, although some have explored its growth in

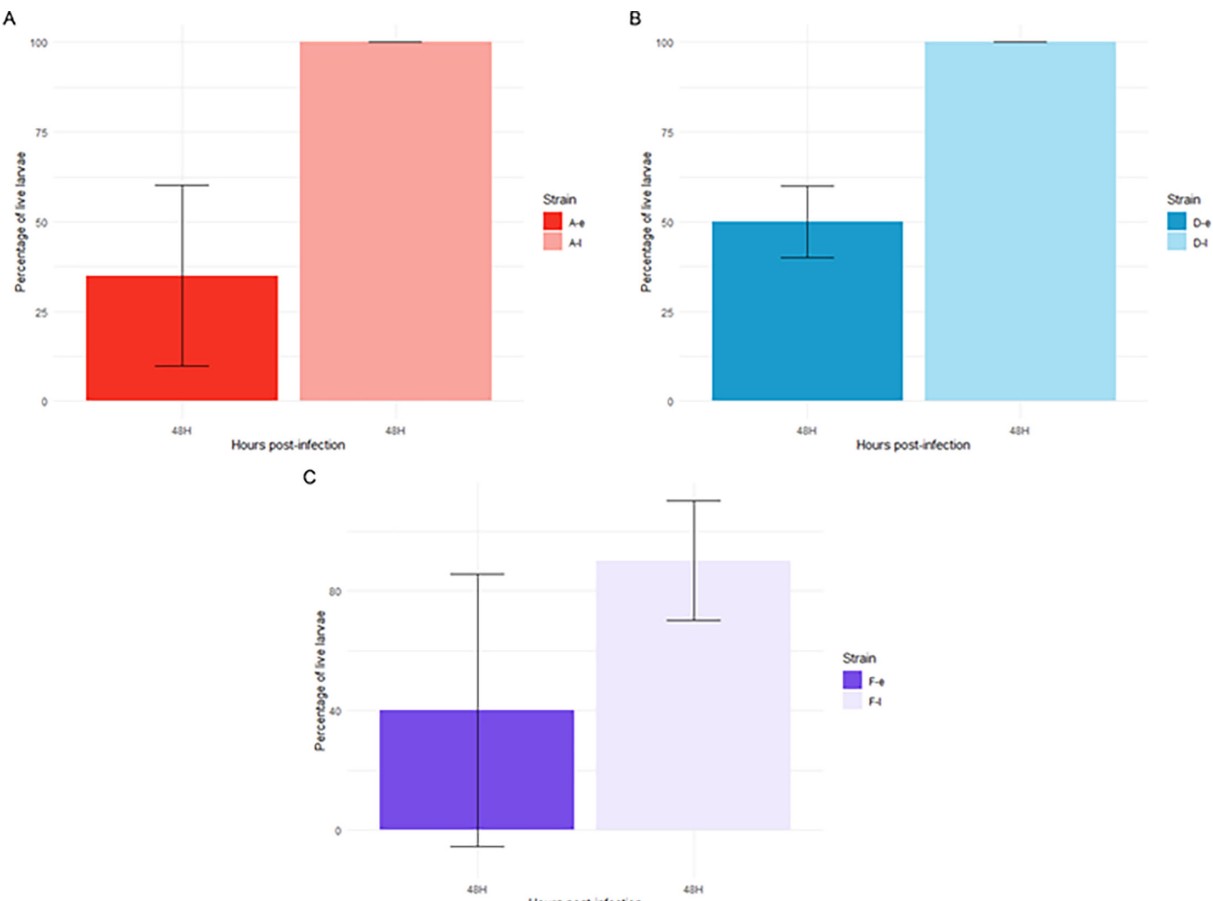

**FIG 6**   Virulence test of early and late isolates from patients A (A), D (B), and F (C) using the *Galleria mellonella* model. Approximately $6 \times 10^2$ CFU were injected into each larva, and 10 larvae were used per isolate. The number of surviving larvae was recorded at 48 h post-infection. Error bars represent the standard deviations from at least three independent experiments.

artificial urine (13, 14) to study its susceptibility to antibiotics (15) and its ability to form biofilm (16). Our study is unique in that it explores the phenotypes of sequential urinary isolates grown in human urine and links these phenotypes to corresponding genomic and proteomic data. In the present study, we studied three different pairs of previously characterized urinary *P. aeruginosa* isolates (early and late) from patients colonized or infected over time by a single clone type (10) to better understand adaptation mechanisms set up by these isolates in the urinary tract. A summary diagram (Fig. 10) grouped the main characteristics observed in this study of *Pseudomonas* isolates adapted to the urinary tract.

Growth of these isolates was compared in different media, including pooled human urine and artificial urine, to evaluate their phenotypic profiles under biologically relevant conditions. Our findings revealed a similar growth reduction of the urinary late isolates from patients D and F, as previously observed in chronic isolates from respiratory or burn wound infections (17, 18). Gordon and colleagues demonstrated that the rapid growth rate of *Escherichia coli* during the early phase of colonization can promote successful establishment and may serve as a virulence factor (19). However, the enhanced growth of the late isolate from patient A relative to the early one in urine was a novel finding not documented in other chronic infections. This observation could be associated with the shorter interval between the initial and subsequent samples for this patient (82 days vs. 113 and 259 days for patients F and D, respectively). In addition, our results showed significant growth variability as well as discrepancy in proteomes in HU and AUM, raising questions about the ability of AUM to mimic HU conditions.

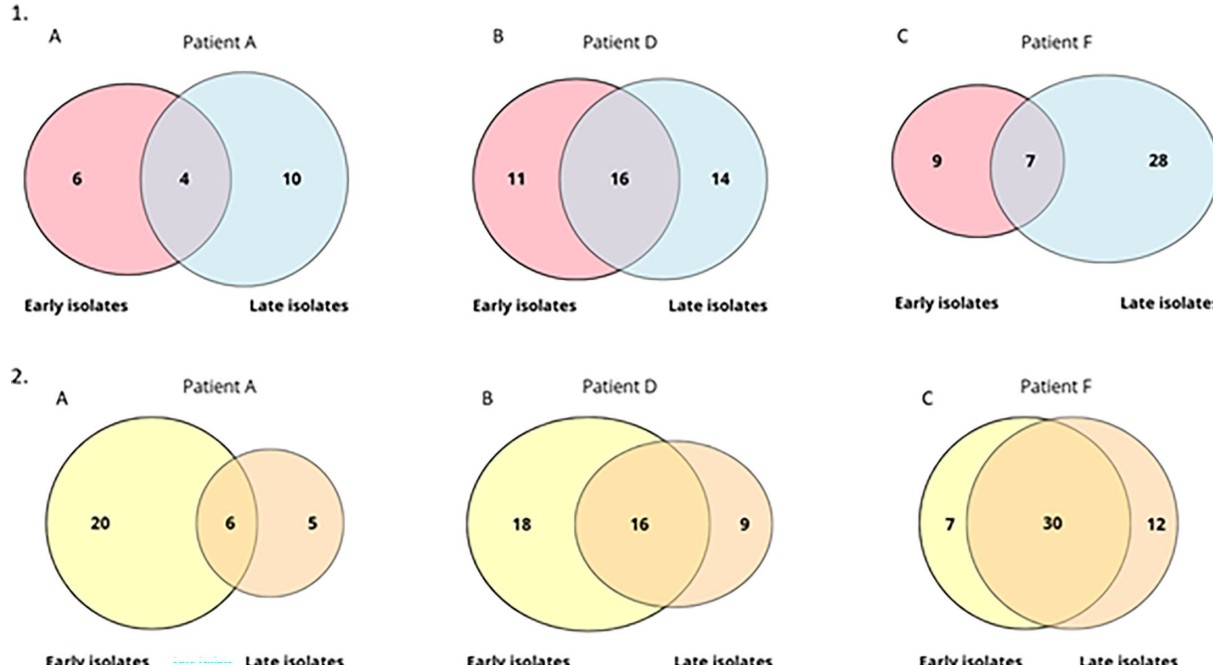

**FIG 7** Venn diagrams representing the number of different metabolic categories corresponding to the overabundant proteins identified in early and late isolates grown in trypticase soy (panel 1) or in human urine (panel 2) from patients A (A), D (B), and F (C). Only proteins with a fold change (FC) >2 were considered.

To further describe the phenotypic characteristics of the late urinary isolates, growth under *in vivo*-relevant stresses was assessed. Indeed, the biochemical characteristics of urine can vary considerably throughout the day, triggering various interesting stresses to be tested. Late isolates from patients D and F appeared more affected by the osmotic stress, particularly at 0.3 and 0.4 M, than those from patient A. Although these NaCl concentrations were higher than typical physiological urine levels (0.1 M [19]), it is noteworthy that the concentration of urea is about 0.3 M in urine (20). This validates the relevance of the concentrations tested and showed that growth of late isolates from two patients was affected at physiological osmotic levels. Nevertheless, Culham and colleagues highlighted that uropathogenic *E. coli* must be urea-resistant and salinity-tolerant to survive in urine and cause infections (21). Despite a higher generation time, our late isolates remained able to cope with high osmotic pressure. This was in line with the overabundance of the glycine betaine transporters OpuCA and OpuCD in the proteome of D-l and F-l late isolates, which are involved in osmoprotectant transport (22, 23).

In addition to high osmolarity, bacteria were subjected to acidic pH stress, which is also encountered in human urine. Urine pH ranges from 5.0 to 8.0 with an average of 6.0 (20). pH tests revealed that acidic conditions slowed the growth of the late isolates more markedly, with doubling times increasing by approximately 35–40%. These results are consistent with those reported in the literature by Mozaheb and colleagues, which described similar, although less marked, outcomes with an 11% increase in doubling time using PAO1 reference strain at pH 5 (24). Surprisingly, exposure to $H_2O_2$ did not significantly modify the growth rates of any isolate. However, proteomic data showed a higher number of oxidoreductase enzymes in the late isolates from the three patients compared to early ones (Table S7), and previous genomic analysis showed mutations or deletions in oxidative-related genes (10). For example, in the A-l isolate, the *katE* and *katN* genes, encoding hydroperoxidase II (HPII) catalase and a non-heme catalase, are deleted, respectively. Nevertheless, it has been described that in *P. aeruginosa,* KatE was not necessary for adaptation to peroxide stress (25), and that KatN of *Salmonella* was not

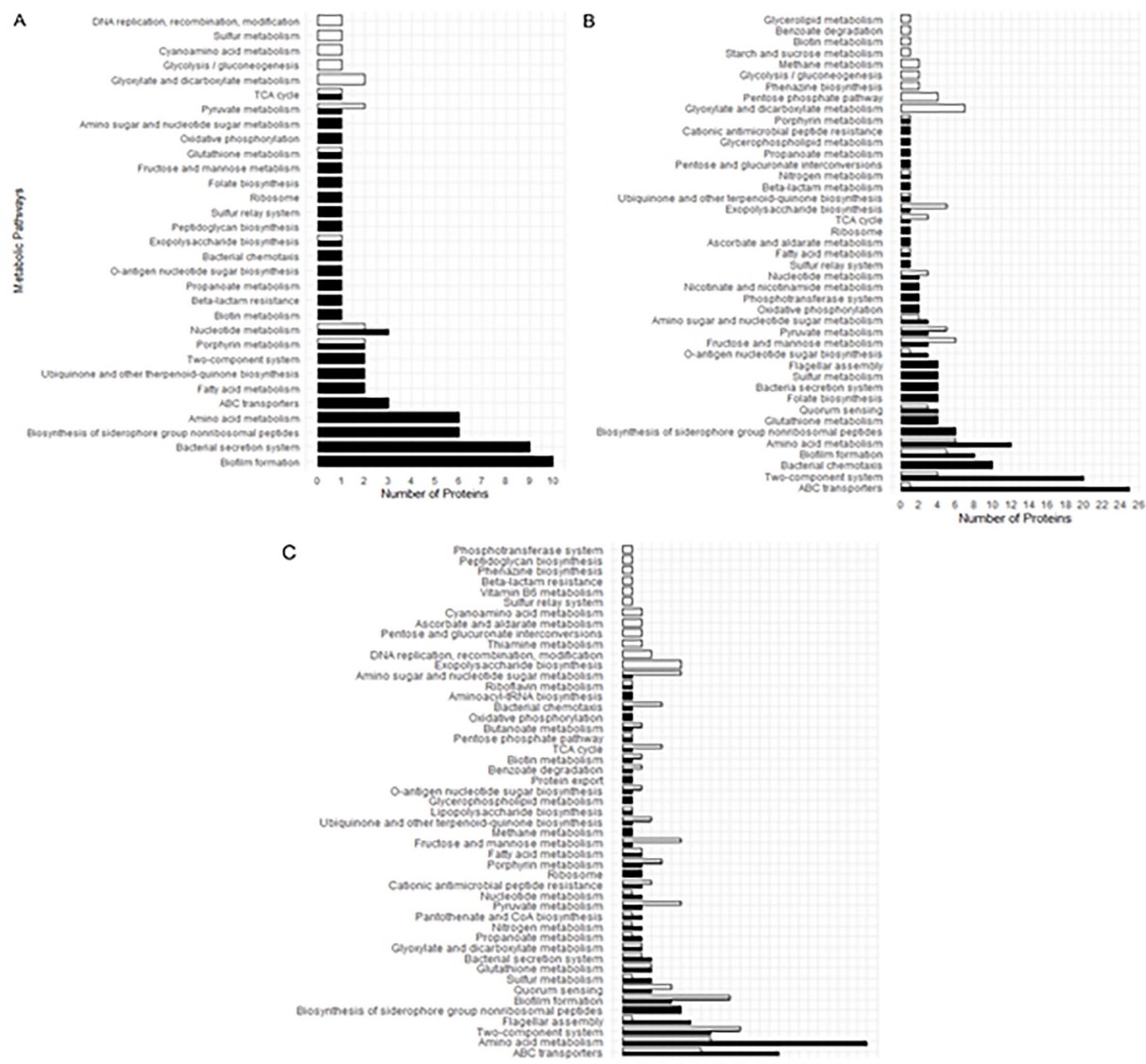

**FIG 8** Number of overabundant proteins classified by functional categories in early (black bars) and in late (white bars) *P. aeruginosa* isolates of patients A (A), D (B), and F (C) grown in HU. Black bars: proteins overabundant in early isolates and consequently underabundant in late isolates. White bars: proteins overabundant in late isolates and consequently underabundant in early isolates. Only proteins with a fold change (FC) >2 were considered.

involved in response to hydrogen peroxide (26). Since oxidative stress is mainly encountered by the bacteria facing immune system, our results suggest that adaptation to this stress is not a requirement for *P. aeruginosa* to persist in urine.

 *P. aeruginosa* chronic infection phenotypes are characterized by downregulation of the virulence factors required to establish acute infections (27, 28). In this study, we used the *G. mellonella* infection model to assess the virulent phenotype of early and late isolates. This model has proven reliable for studying the pathogenesis of numerous human pathogens because the innate immune systems of *Galleria* larvae and mammals show a high degree of structural and functional homology. In contrast to studies of host-pathogen interactions carried out in urinary cell culture models or organoids, these experiments assess the ability of bacteria to cope with host defenses such as enzymes and reactive oxygen species after phagocytosis in hemocytes and antimicrobial peptides (29). We showed that late isolates were less virulent than early isolates in the *G. mellonella* model, which can be explained, at least in part, by various deletions and mutations

**TABLE 1** Number of overabundant proteins involved in the biosynthesis of the two main siderophores of *P. aeruginosa* (pyochelin and pyoverdine) as well as in flagellum and chemotaxis, when early and late isolates from patients A, D, and F were grown in human urine (HU) or trypticase soy (TS)[a]

| | Number of overabundant proteins | |
|---|---|---|
| Isolates | Siderophores | Flagellum and chemotaxis |
| Patient A | | |
| Early in TS | 0 | 1 |
| Late in TS | 2 | 1 |
| Early in HU | 13 | 1 |
| Late in HU | **0** | 0 |
| Patient D | | |
| Early in TS | 0 | 13 |
| Late in TS | 1 | 0 |
| Early in HU | 14 | 14 |
| Late in HU | **0** | 0 |
| Patient F | | |
| Early in TS | 0 | 13 |
| Late in TS | 2 | 0 |
| Early in HU | 13 | 8 |
| Late in HU | **0** | 4 |

[a]Bolded values indicate the absence of overabundant proteins in the three late isolates grown in HU compared to early isolates.

in key virulence genes. For instance, late isolates from patient A exhibited deletion of the *hcn* operon involved in hydrogen cyanide production; the *exoY* gene, encoding an exotoxin secreted by T3SS; three TonB-dependent receptor genes involved in the import of siderophore-chelated iron; and an extracellular factor (ECF) sigma factor of the RNA polymerase (10). Interestingly, deletions in the same genes were also observed in a Belgian epidemic CF isolate of *P. aeruginosa* (30), suggesting a selective advantage in specific environments such as the lungs of CF patients or the urinary tract during chronic infection/colonization. In late isolates from patient D, a nonsense mutation in the *exsA* gene, which is a transcriptional activator of the T3SS, as well as mutations and deletions in various motility-related genes were found (10). This is in agreement with mutations in genes encoding transcriptional regulators like *exsA* and flagellum components frequently reported in CF isolates during host adaptation, marking the transition from acute to chronic infection phenotypes (28, 31, 32). Finally, deletion of the *fap* operon, which contributes to the pathogenesis and biofilm formation in *P. aeruginosa* (33, 34), was observed in late isolates from patient F.

In this study, we used a proteomic approach to better understand the adaptive evolution of early and late *P. aeruginosa* urinary clinical isolates at the level of protein expression. Except for proteins corresponding to genes deleted in the late isolates, comparison of early and late isolates revealed an enrichment of enzymes linked to diverse metabolic pathways related to energy metabolism in all late isolates, a feature previously described in the literature (16, 18, 35, 36). Tielen and colleagues showed an adaptation of the central metabolism of urinary *P. aeruginosa* isolates, especially with the activation of the glyoxylate bypass. Some studies also revealed that the reprogramming of metabolic pathways, such as pyruvate metabolism, the TCA cycle, and glyoxylate shunt, leads to a host-specialized metabolism that plays a key role in achieving persistence together with a lower virulence (14, 35–37). These observations are in agreement with the proteomic data, where an overabundance of enzymes involved in the glyoxylate cycle and pyruvate metabolism was identified in late isolates adapted to the specific composition of urine. Finally, proteomic profiles revealed that fewer metabolic pathways were expressed in late isolates from patient A compared to those from patients D and F. This may be correlated with the longer colonization period for strains D and F, which resulted in longer selection pressure. Of note, RT-qPCRs

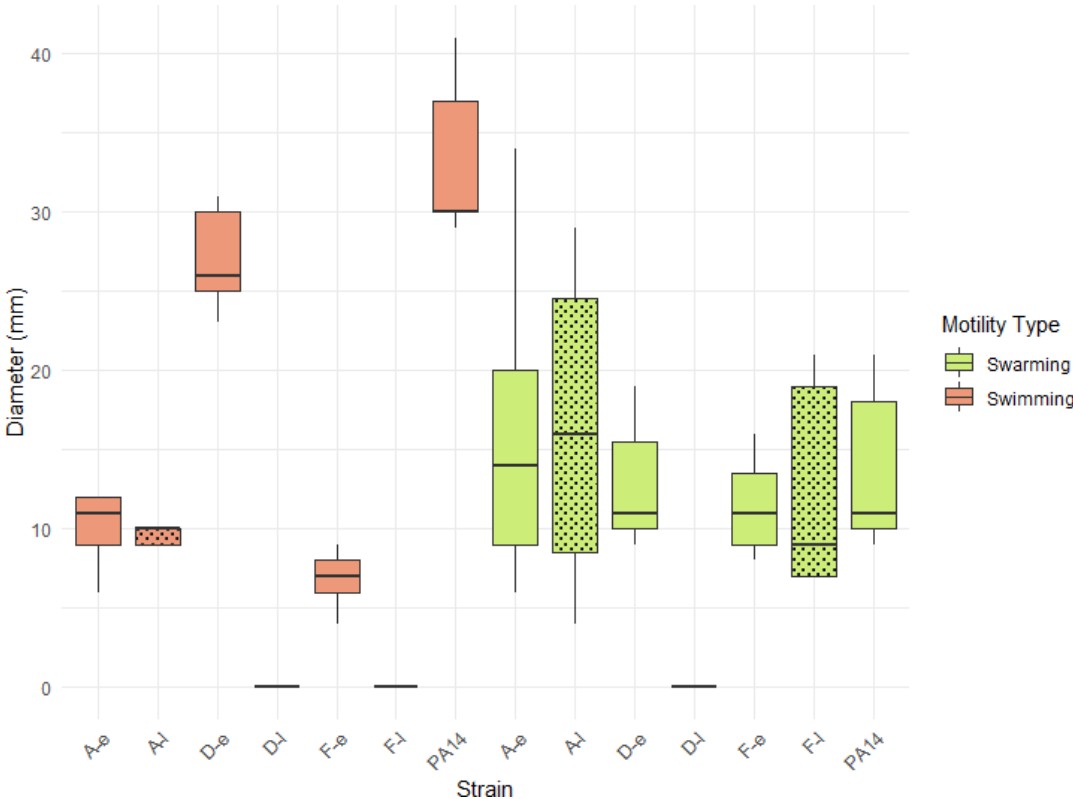

**FIG 9** Swimming and swarming motility diameters of early and late isolates from patients A, D, and F. PA14 strain was used as a positive control. Boxplots represent the median and interquartile range for the swimming (orange) and swarming (green) diameters of the different strains. Error bars represent the standard deviations from at least three independent experiments. Statistical significance was determined as $P < 0.05$ using a two-sample Student's $t$-test for independent samples.

performed on two genes (*pchD* and *sodM*) encoding proteins found in the proteomic data suggest their transcriptional regulation (data not shown). These genes were found to be downregulated in late isolates: *pchD* by 29-, 13-, and 38-fold in patients A, D, and F, respectively; and *sodM* by 1082-, 43-, and 67-fold in patients A, D, and F, respectively, compared to those in early strains.

In contrast with proteomic studies on *P. aeruginosa* under CF conditions (38, 39), which reported either an increase or no change in the number of proteins involved in iron metabolism, it is interesting to note the absence of overabundant proteins involved in siderophores biosynthesis or iron-related proteins in late isolates grown in HU. These findings were unexpected, as urine is an iron-limited medium (20) that typically favors the production of such proteins. This absence of iron-dependent regulation may represent an adaptive feature specific to urinary isolates. Based on these data, siderophores could be promising biomarkers for the diagnosis of difficult-to-treat *P. aeruginosa* chronic UTIs.

The near-total absence of overabundant flagellum and chemotaxis-related proteins in late isolates compared to early ones for two of the three patients studied led us

**TABLE 2** Summary of phenotypic comparisons between early and late isolates for the three patients[a]

| | | | | | Growth under stresses | | | | |
|---|---|---|---|---|---|---|---|---|---|
| | TS growth | HU growth | AUM growth | NaCl | pH | H$_2$O$_2$ | Virulence | Swimming | Swarming |
| Patient A | ↗ | ↘ | ↘ | ↗ | ↗ (except pH 5.5 ↘) | = | ↘ | = | = |
| Patient D | ↗ | ↗ | ↗ | ↗ | ↗ | = | ↘ | ↘ | ↘ |
| Patient F | ↗ | ↗ | ↗ | ↗ | ↗ | = | ↘ | ↘ | = |

[a]Arrows indicate changes in the behavior in the late isolate compared with its early counterpart. equal sign means no change.

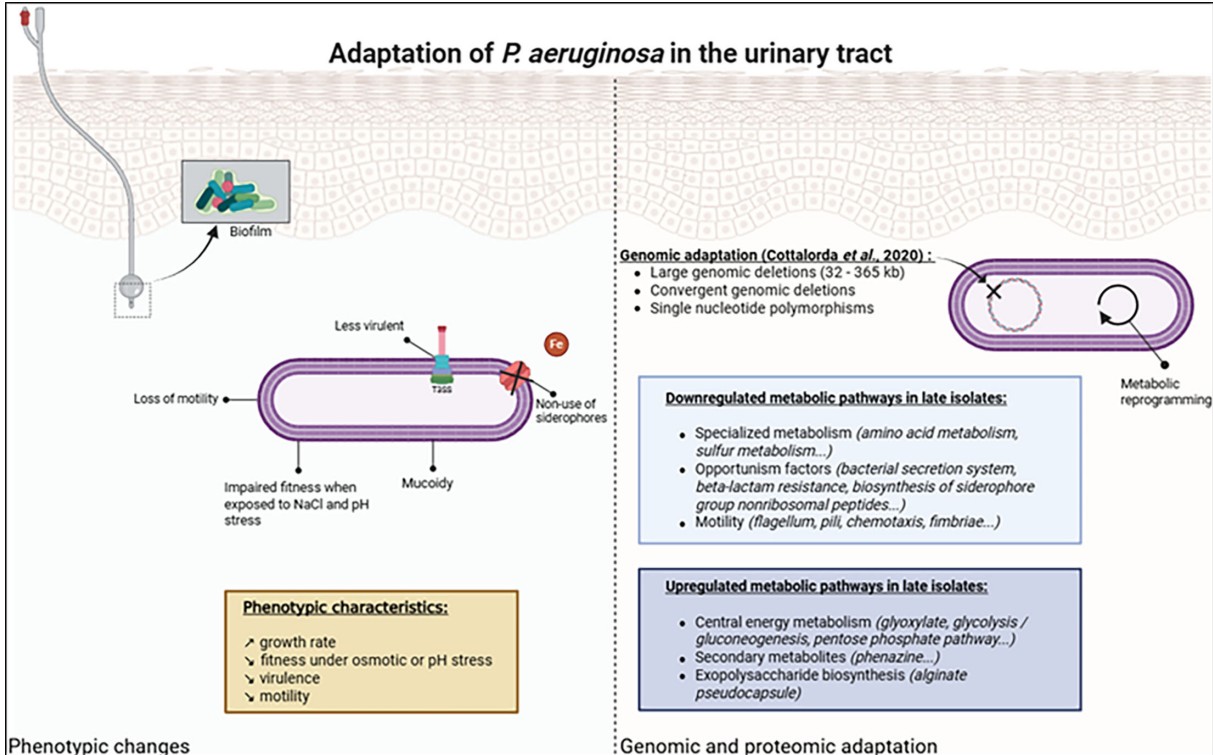

**FIG 10** Main characteristics observed in late urinary isolates of *P. aeruginosa* during their adaptation to the urinary tract. Illustrations created with Biorender.com

to examine their swimming and swarming abilities. The loss of swimming exhibited by the late isolates of patients D and F is in accordance with findings reported in the literature for chronic *P. aeruginosa* infections in CF airways (40–42), burn wounds (17), and urinary tract infections (43). The loss of the swimming in F-l isolate may be related to genomic alteration (10). This isolate carried a nonsense mutation in the *mucA* gene (10), which encodes an anti-sigma factor that sequesters the alternative sigma factor AlgU acting as a negative regulator of flagella-associated motility (44). Moreover, Pulcrano and co-workers demonstrated that *mucA* mutants exhibit a lower motility than non-mutated isolates of *P. aeruginosa* (45). For the D-l isolate, the high number of SNPs or deletions in various motility- or chemotaxis-related genes (10) could explain its non-swimming and non-swarming phenotypes. Overall, our phenotypic results align with findings observed during respiratory infections in CF patients (42), which may reflect an adaptation convergence between our urinary isolates and strains adapted to the CF lung environment, but these results need to be confirmed on a larger cohort of sequential isolates. Despite this, the use of three pairs of isolates from three different patients enabled us to identify common features related to persistence in the urinary tract that can be further validated on new samples. By integrating analyses of large genomic deletions and SNPs with proteomic data and phenotypic results, we have gained a deeper understanding of the adaptation mechanisms of our isolates persisting in the urinary tract, providing valuable insights into the processes underlying *P. aeruginosa* UTIs. Further additional global approaches (metabolomic and transcriptomic) should provide more information about the complex network leading to the persistence of *P. aeruginosa* into the urinary tract.

## MATERIAL AND METHODS

### Bacterial isolates and growth conditions

For this study, we used a set of six *P. aeruginosa* urinary isolates collected sequentially from three patients (A, D, and F) at the Rouen University Hospital who were previously characterized (10). Briefly, patients A and D were immunocompromised, and patients D and F had urinary comorbidity. The three patients received at least one course of antibiotic therapy in the six months preceding the first urine culture, and patient A also received antibiotic therapy between two urine samples. This set included one early isolate (from the first sample collected) and one late isolate (characterized by large genomic deletions) per patient. The nomenclature of each isolate was as follows: the letters (A-D-F) identified the patients and the small letters (e and l) corresponded to the early or late urine isolate, respectively (Fig. 1).

*P. aeruginosa* isolates were grown in the following media: trypticase soy broth (TS), pooled human urine (HU) (filtered through 0.2 µm filter, collected from adults without urinary tract abnormalities and without any prescription medication taken in the last two weeks) (BioIVT, West Sussex, UK), and artificial urine medium (AUM) prepared according to Brooks and Keevil's protocol (13) with minor modifications. Briefly, all the 16 compounds necessary for the AUM were mixed, and the pH was adjusted to 6.5. To limit precipitate formation, 2-(N-morpholino)-ethanesulfonic acid (MES buffer) was added at a final concentration of 100 mM. For growth curve assessments, isolates were grown overnight in TS at 37°C with shaking at 150 rpm and normalized to an optical density ($OD_{600nm}$) of 0.01 in 1 mL of TS, HU, or AUM. A 200 µL of each standardized suspension was inoculated in triplicate in a microtiter plate. The plate was then placed inside a humidity cassette and incubated at 37°C for 24 h under continuous double orbital shaking at 108 rpm in a Spark microplate reader (Tecan, Männedorf, Switzerland). $OD_{600nm}$ was measured every 15 min, and all experiments were performed at least three times. Cultures in TS were used as positive control and fresh medium without bacteria as negative control for each experiment. Statistical analyses were performed using R software (v 4.4.1, R foundation, Vienna, Austria, HYPERLINK "https://www.r-project.org/"https://www.R-project.org/). Generation times were determined, and statistical analyses were performed by an independent *t* test on R software (v.4.4.1). Differences with a *P* value of less than 0.05 were considered statistically significant.

### Stress conditions

Isolates were grown overnight in TS medium at 37°C with shaking at 150 rpm. Then, 500 µL of these cultures were centrifuged (5000 rpm and 5 min), and the resulting pellets were resuspended in TS supplemented with different NaCl concentrations (0, 0.1, 0.2, 0.3, and 0.4 M) (NaCl concentration TS ~0.08M), $H_2O_2$ concentrations (0, 0.1, 0.2, 0.3, and 0.4 mM), or adjusted to different pH values (6, 5.5, and 5) (pH TS ~7) without adding buffer. $OD_{600nm}$ was adjusted to 0.01, and growth was recorded every 15 min for 24 h using the Spark microplate reader. Each experiment was repeated at least three times. *P* values were determined using independent *t*-tests.

### *Galleria mellonella* infection assays

*G. mellonella* larvae were infected with *P. aeruginosa* cells as previously described (46). Briefly, using a syringe pump (kdScientific Inc., Massachusetts, USA), larvae (approximately 0.3 g and 3 cm in length) were infected subcutaneously with washed *P. aeruginosa* isolates from an overnight culture in TS, with an inoculum of $6 \times 10^2$ CFU per larva administered in 10 µL of saline buffer. For each test, ten insects were infected per isolate, and the experiments were repeated at least three times. Five larvae injected with saline buffer were used as negative controls. Larval survival was then monitored at 48 h post-infection. The results were analyzed by an independent *t* test, and comparisons with *P* value of less than 0.05 were considered statistically significant.

## Sample preparation and proteomic analysis by mass spectrometry

Proteins were extracted from *P. aeruginosa* cells grown in 20 mL of TS, HU, or AUM harvested in the late-exponential phase. Following incubation of the cell pellets at −80°C, cells were resuspended in 2 mL of recovery buffer (Tris HCl 50 mM [pH 7.5] + $Na_2SO_4$ 50 mM+Glycerol 15%) and lysed using MN bead tubes type B (Macherey-Nagel Inc., Pennsylvania, USA) with a FastPrep instrument (MP Biomedical LLC, Santa Ana, CA, USA) for 2 min 30 s. The lysates were then centrifuged for 10 min at 10,000 × *g* at 4°C to remove cell debris. Quantification of total proteins was performed using the Pierce BCA Protein Assay Kit (ThermoScientific, Massachusetts, USA) according to the manufacturer's instructions. Five micrograms (µg) of each protein extract were digested with trypsin/Lys-C overnight at 37°C. After protein or peptide desalting and concentration, chromatography was performed using a NanoElute ultrahigh-pressure Nano flow chromatography system (Bruker Daltonics, Billerica, MA, USA). Mass spectrometry experiments were carried out on a TIMS-TOF Pro mass spectrometer (Bruker Daltonics), and spectra were acquired in positive mode over a mass range of 100–1700 m/z. These data were processed with MaxQuant version 1.6.7.0, and (MS)/MS spectra were searched with the Andromeda search engine against the Uniprot *P. aeruginosa* database. Bioinformatic analysis and data visualization were performed using Perseus. Comparisons were carried out between early and late isolates from the three patients across the three media (TS, AUM, and HU). Three sample tests were performed using Student's t test with a permutation-based false discovery rate (FDR) of 0.05. Values of log2 fold change (Log2FC) less than −2 or greater than 2, with a corrected *P* value less than 0.05, were considered to be statistically less or more abundant, respectively.

## Motility assays

Swimming and swarming motility assays were performed as previously described, with slight modifications (47). Swimming motility was assessed using nutrient agar plates containing 0.3% agar (w/v) (BD Difco TM, New Jersey, USA), 5 g/L NaCl (VWR chemicals, Pennsylvania, USA), and 10 g/L tryptone (ThermoScientific). Swarming motility was tested on nutrient agar plates containing 0.5% agar (w/v), 2.2 g/L NaCl, 1.3 g/L yeast extract (ThermoScientific), 4.5 g/L tryptone, and 5 g/L D(+) glucose (Sigma-Aldrich, Missouri, USA). Cultures grown in TS medium were diluted to an $OD_{600}$ of 0.05 and incubated at 37°C with shaking at 150 rpm until reaching an $OD_{600}$ of 0.4–0.5. Swim plates were inoculated with a sterile needle and incubated for 24 h at 37°C. For the swarming assay, 1 µL of bacterial suspension was spotted onto the surface of the agar plates and incubated for 24 h at 37°C. Motilities were then assessed by measuring the diameters of radial growth on the plates. Each experiment was repeated three times, and *P* values were determined using independent *t* tests.

## ACKNOWLEDGMENTS

We are grateful to the University of Rouen Normandy for funding the cursus of C. Martin-Duval.

The authors warmly thank Stéphanie Legris and Mamadou Godet for technical assistance.

## AUTHOR AFFILIATIONS

[1]Univ Rouen Normandie, Université de Caen Normandie, INSERM, Normandie Univ, DYNAMICURE UMR 1311, Rouen, France

[2]Department of Bacteriology, Univ Rouen Normandie, Université de Caen Normandie, INSERM, Normandie Univ, DYNAMICURE UMR 1311, Rouen, France

[3]Université de Caen Normandie, Univ Rouen Normandie, INSERM, Normandie Univ, DYNAMICURE UMR 1311, Caen, France

[4]Plateforme Proteogen SFR ICORE 4206, Université de Caen Normandie, Caen, France

## AUTHOR ORCIDs

Caroline Martin-Duval ⬤ http://orcid.org/0009-0003-1678-0734
Sandrine Dahyot ⬤ http://orcid.org/0000-0002-4230-1886
Jean-Christophe Giard ⬤ http://orcid.org/0000-0001-8588-2732

## AUTHOR CONTRIBUTIONS

Caroline Martin-Duval, Conceptualization, Investigation, Writing – original draft, Writing – review and editing | Sandrine Dahyot, Funding acquisition, Methodology, Supervision, Writing – review and editing | Inès Coquisart, Investigation | Benoit Bernay, Formal analysis, Methodology | Martine Pestel-Caron, Conceptualization, Funding acquisition, Methodology, Supervision, Validation, Writing – review and editing | Jean-Christophe Giard, Conceptualization, Investigation, Methodology, Supervision, Validation, Writing – original draft, Writing – review and editing

## DATA AVAILABILITY

The data sets presented in this study can be found in online repositories. The names of the repository/repositories and accession number(s) can be found below: PRJNA656414. The LC-MS/MS proteomics data have been deposited to the ProteomeXchange Consortium via the PRIDE partner repository with the data set identifier IPX0010955001.

## ETHICS APPROVAL

According to the French regulation on observational database analyses, the isolate collections used in this study did not need specific informed consent requirements; it was registered by the Directorate of Clinical Research and Innovation of the Rouen University Hospital Center under the number 2018/413/OB.

## ADDITIONAL FILES

The following material is available online.

### Supplemental Material

**Supplemental material (Spectrum00456-25-S0001.docx).** Tables S1 to S6; Fig. S1 to S3.
**Table S7 (Spectrum00456-25-S0002.xlsx).** Proteins significantly overabundant and underabundant of growing cell (OD of 1) of *P. aeruginosa* strains grown in TS, AUM, and HU.

### Open Peer Review

**PEER REVIEW HISTORY (review-history.pdf).** An accounting of the reviewer comments and feedback.

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
