## [Reviewer comments · Microbiology Spectrum]

Microbiology Spectrum

Insights on pathoadaptation of sequential *Pseudomonas aeruginosa* isolates to the urinary tract

Caroline Martin-Duval, Sandrine Dahyot, Inès Coquisart, Benoit Bernay, Martine Pestel-Caron, and Jean-Christophe Giard

Corresponding Author(s): Caroline Martin-Duval, Universite de Rouen Normandie

Review Timeline:

Submission Date:	February 14, 2025
Editorial Decision:	May 5, 2025
Revision Received:	June 16, 2025
Accepted:	June 26, 2025

Editor: Estela Galvan

Reviewer(s): Disclosure of reviewer identity is with reference to reviewer comments included in decision letter(s). The following individuals involved in review of your submission have agreed to reveal their identity: Sébastien Bontemps-Gallo (Reviewer #2)

Transaction Report:

DOI: <https://doi.org/10.1128/spectrum.00456-25>

Re: Spectrum00456-25 (**Insights on pathoadaptation of sequential *Pseudomonas aeruginosa* isolates to the urinary tract**)

Dear Mrs. Caroline Martin-Duval:

Thank you for the privilege of reviewing your work. Below you will find my comments, instructions from the Spectrum editorial office, and the reviewer comments.

Revision Guidelines

Sincerely,
Estela Galvan
Editor
Microbiology Spectrum

Reviewer #1 (Comments for the Author):

The present study provides a comprehensive investigation of *Pseudomonas aeruginosa* adaptation in the urinary tract, a scenario that remains understudied in comparison to respiratory infections such as those in cystic fibrosis patients. By integrating genomic, phenotypic, and proteomic analyses, the authors offer new insights into the pathoadaptive mechanisms of sequential isolates collected from patients with recurrent urinary tract infections (UTIs). Their findings contribute to a better understanding of

microbial persistence, virulence attenuation, and metabolic shifts in *P. aeruginosa* during infection.

The study effectively integrates genomics, proteomics, and virulence assays, providing a comprehensive view of bacterial adaptation. However, most of the results presented were not statistically significant, making the manuscript weaker when come to conclusions. Many comparisons (growth, stress responses, virulence, and proteomics) did not reach statistical significance, making it difficult to draw firm conclusions.

Major changes and suggestions

- 1) Did the authors perform transcriptomic analysis? While proteomic data provide excellent functional insights, RNA sequencing (RNA-seq) could have helped link gene expression changes to observed phenotypes. Transcriptomic analysis is advised and will improve the quality of the paper.
- 2) The study identifies key proteomic and genomic changes but does not experimentally validate their role in adaptation. Gene knockouts or complementation studies could strengthen the conclusions. Please include additional validation experiments. Studies could confirm whether flagellum loss, siderophore reduction, or virulence attenuation are direct adaptive mechanisms.
- 3) Lack of Longitudinal Sampling Beyond Two Time Points. The authors compare early and late isolates, but intermediate isolates could provide more granular insights into the stepwise adaptation of *P. aeruginosa*. Were any intermediate isolates recovered?
- 4) Clarify patient variability: Why do some isolates increase growth rates while others decline? Could it be due to different treatment histories or host factors?

The study provides valuable insights into bacterial adaptation, but the weak statistical significance of many results reduces the strength of its conclusions. A more cautious interpretation, additional statistical validation, and possibly increasing the sample size would make the manuscript stronger.

Reviewer #2 (Comments for the Author):

Summary

This study investigates the adaptive evolution of *Pseudomonas aeruginosa* during chronic urinary tract colonization by analyzing three pairs of sequential urinary isolates (early vs. late) collected from different patients. Late isolates exhibited diverse phenotypic adaptations, including reduced growth in human urine (HU) and artificial urine medium (AUM) in two patients, and enhanced growth in one. When exposed to osmotic and acidic stresses, late isolates generally showed impaired fitness, while oxidative stress had no notable impact. Proteomic analyses revealed significant shifts in protein abundance: late isolates were enriched in energy metabolism-associated proteins, and depleted in virulence-associated proteins such as siderophore biosynthesis components and flagellar motility factors. Functional assays confirmed reduced virulence and motility in most late isolates. These findings point to a transition from acute virulence to persistence, likely driven by host-specific selective pressures. Importantly, the study demonstrates that AUM poorly mimics the proteomic landscape observed in HU, highlighting the need for physiologically relevant media in such investigations. While some patient-specific differences were observed, the convergence of proteomic and phenotypic trends suggests common adaptive trajectories in the urinary environment.

Major Concerns:

- 1/ (lines 156-167, Fig. 6) *P. aeruginosa* is a highly versatile pathogen with a broad host range. However, given that this study focuses on urinary tract colonization, the biological relevance of using *Galleria mellonella* as a virulence model requires explanation. How well does it recapitulate key host-pathogen interactions in the urinary tract?
- 2/ (lines 114-116) Why were the stress assays performed in TS medium rather than in AUM or HU? Do the authors have data on the initial pH and osmolarity of the media used? In healthy individuals, urinary osmolarity can vary considerably throughout the day-even within a single patient-depending on factors such as hydration status, ambient temperature, humidity, and physical activity. Similarly, urine pH ranges from 4 to 8, with a tendency toward acidity, and is influenced by factors like diet. These physiological variations should be taken into consideration when selecting media for stress response experiments.
- 3/ To improve reader comprehension, it would be helpful to include a comparative schematic summarizing the main proteomic findings and highlighting the most notable differences or similarities across the six clinical isolates. For instance, in Figure 7, are the differentially abundant proteins conserved across patients? What is the relative impact of TS versus HU medium on protein expression?
In the same vein, including a summary table or graphic of the phenotypic test results for each of the six isolates, and comparing their progression over time, would significantly enhance the clarity of the discussion. Although the authors describe differentially expressed proteins under various conditions, it remains challenging to identify specific systems or pathways that should be explored in more detail. A clearer synthesis would strengthen the manuscript.

Minor Concerns

Lines 50-53: What is the actual prevalence of *P. aeruginosa* in nosocomial UTIs? Including precise epidemiological data would be useful.

Lines 58-59: The statement that adaptation mechanisms have been "already" well studied in CF patients should be softened, as these are still under investigation.

Lines 70-73 / Fig. 1: Please provide relevant patient details (e.g., age, comorbidities, site of infection, antibiotic use between isolates), which may influence evolutionary outcomes.

Lines 93-94: Reintroduce and explain the rationale for using HU and AUM media.

Lines 114-116 / Fig. 3: Have the authors measured the initial osmolarity of TS? Why not use AUM, which would better represent urinary osmotic conditions?

Lines 402-403: Please clarify how AUM was prepared, including modifications to the original protocol.

Fig. 2: Generation time is reported-was final biomass (yield) also assessed?

Line 420: Was a buffering agent used to adjust pH? Was pH monitored throughout the culture period?

Revised version of the manuscript « Insights on pathoadaptation of sequential *Pseudomonas aeruginosa* isolates to the urinary tract »

Reference: Spectrum00456-25

We would like to thank the Editor and Reviewers for their helpful comments concerning our manuscript. We are convinced that these comments have contributed to the overall improvement of our paper.

Reviewer #1 (Comments for the Author):

The present study provides a comprehensive investigation of *Pseudomonas aeruginosa* adaptation in the urinary tract, a scenario that remains understudied in comparison to respiratory infections such as those in cystic fibrosis patients. By integrating genomic, phenotypic, and proteomic analyses, the authors offer new insights into the pathoadaptive mechanisms of sequential isolates collected from patients with recurrent urinary tract infections (UTIs). Their findings contribute to a better understanding of microbial persistence, virulence attenuation, and metabolic shifts in *P. aeruginosa* during infection.

The study effectively integrates genomics, proteomics, and virulence assays, providing a comprehensive view of bacterial adaptation. However, most of the results presented were not statistically significant, making the manuscript weaker when come to conclusions. Many comparisons (growth, stress responses, virulence, and proteomics) did not reach statistical significance, making it difficult to draw firm conclusions.

We thank the reviewer for this comment but it doesn't seem entirely accurate to say that many comparisons do not show significant differences. For example, our statistical analyses revealed that the A-I strain grew significantly faster in HU and AUM compared to A-e (L102-104 of the revised manuscript), or that late isolates showed significantly impaired growth when confronted to acid and osmotic stresses compared to early counterparts (L125-128 and L138-141 of the revised manuscript). Furthermore, we have clearly demonstrated that the late isolates were significantly less virulent in the *Galleria mellonella* infection model (L165-167 of the revised manuscript).

In the initial version of the manuscript, not all p-values were explicitly reported in the text, which may have hampered the assessment of the robustness of the data. We therefore took this comment into account and added p-values for experiments with significant results (L104, L132, L140-141 of the revised manuscript).

Major changes and suggestions

1) Did the authors perform transcriptomic analysis? While proteomic data provide excellent functional insights, RNA sequencing (RNA-seq) could have helped link gene expression changes to observed phenotypes. Transcriptomic analysis is advised and will improve the quality of the paper.

We agree that this type of approach would be of great interest to better understand the regulatory mechanisms involved in the persistence of *Pseudomonas* in the urinary tract. Therefore, as suggested, RNAseq studies will be an important part of our future experiments.

RT-qPCRs were nevertheless performed to assess the expression of two genes encoding proteins highlighted in the proteomic data (*pchD* and *sodM*). These data suggest transcriptional regulation of these genes in late isolates since their expression were repressed 29, 13 and 38-fold for the *pchD* gene in patients A, D and F respectively and 1082, 43 and 67-fold for the *sodM* gene in patients A, D and F respectively, compared to those of early strains. This was added in the discussion section of the revised version of the manuscript (L382-386).

2) The study identifies key proteomic and genomic changes but does not experimentally validate their role in adaptation. Gene knockouts or complementation studies could strengthen the conclusions. Please include additional validation experiments. Studies could confirm whether flagellum loss, siderophore reduction, or virulence attenuation are direct adaptive mechanisms.

We thank the reviewer for this suggestion but the adaptation process appears to result from the expression of many factors. This is why it seemed relevant to us to carry out a global proteomic analysis rather than a study of specific genes by mutagenesis and/or complementation. In this study, the "validation" of our observations was based on the phenotypic behaviors of the cells (*ie.* in terms of mobility and stress response) or on already well-established knowledge of this bacterium as described in the discussion (*ie.* virulent factors [L342-343 of the revised manuscript], siderophores roles [L387-390 of the revised manuscript]). In this context, as mentioned in the previous comment, future global transcriptomic data will provide additional information on the adaptation of *Pseudomonas* to the urinary tract.

3) Lack of Longitudinal Sampling Beyond Two Time Points. The authors compare early and late isolates, but intermediate isolates could provide more granular insights into the stepwise adaptation of *P. aeruginosa*. Were any intermediate isolates recovered?

We agree that intermediate isolates could have provided more information about the adaptive trajectory of *P. aeruginosa*. Unfortunately, we did not recover any intermediate isolates from our patients, except for the patient A but with only a 14-day interval between the first and the second samples and, this isolate did not have genetic deletions. However, the use of three pairs of isolates from three different patients allowed us to identify common characteristics linked to persistence in

the urinary tract which can be further tested on a new collection of longitudinal samples. This comment has been added to the discussion of the revised version of the manuscript (L410-412).

4) Clarify patient variability: Why do some isolates increase growth rates while others decline? Could it be due to different treatment histories or host factors?

Thank you for this relevant remark. Indeed, it is very likely that patient's history and host factors may influence the phenotypic behavior of isolates. The characteristics of the patients from whom these isolates were obtained have been published previously (Cottalorda et al., 2021, Front Microbiol, DOI: 10.3389/fmicb.2020.611246). Following the reviewer's comment, we have included in the revised version key metadata concerning patient history and antibiotic treatment history before or between isolate collection. We recalled that patients A and D were immunocompromised, that patients D and F had urinary comorbidity, and that patients A, D and F had received at least one course of antibiotics in the six months prior to collection of the first urine sample or in the period between the two samples studied (L422-425 of the revised manuscript).

Based on these elements we hypothesize that the higher growth rate in urine of the late isolate of patient A than those of patients D and F could be associated with the shorter interval between the initial and subsequent samples for this patient (82 days vs 113 and 259 days for patients F and D, respectively) as mentioned in L307-309 of the revised manuscript.

Reviewer #2 (Comments for the Author):

Major Concerns:

1/ (lines 156-167, Fig. 6) *P. aeruginosa* is a highly versatile pathogen with a broad host range. However, given that this study focuses on urinary tract colonization, the biological relevance of using *Galleria mellonella* as a virulence model requires explanation. How well does it recapitulate key host-pathogen interactions in the urinary tract?

Thanks for this comment. The main objective of this work was to identify the phenotypic and proteomic characteristics of *P. aeruginosa* strains adapted to the urinary tract, which were either responsible for urinary tract infection or colonization. In this perspective, the use of the *Galleria mellonella* larvae infection model offered the advantage of assessing the ability of strains to cope with the innate immune response and thus to evaluate their pathogenicity. Indeed, this model has emerged as a reliable model to study the pathogenesis of many human pathogens and the innate immune systems of *Galleria* larvae and mammals share a high degree of structural and functional homology (Ménard et al., 2021, Front. Cell. Infect. Microbiol., DOI: 10.3389/fcimb.2021.782733). Importantly, killing of pathogens occurs in the same way as in mammals, *i.e.*, through the activity of enzymes and reactive oxygen species after phagocytosis in haemocytes and by production of antimicrobial peptides. The study of host-pathogen interactions could be considered through

experiments on urinary cell cultures or organoids. These insights and perspectives have been added to the discussion section of the revised manuscript (L343-350).

2/ (lines 114-116) Why were the stress assays performed in TS medium rather than in AUM or HU? Do the authors have data on the initial pH and osmolarity of the media used? In healthy individuals, urinary osmolarity can vary considerably throughout the day-even within a single patient-depending on factors such as hydration status, ambient temperature, humidity, and physical activity. Similarly, urine pH ranges from 4 to 8, with a tendency toward acidity, and is influenced by factors like diet. These physiological variations should be taken into consideration when selecting media for stress response experiments.

As the reviewer rightly points out, the biochemical characteristics of urine can vary and trigger stresses for the bacteria present. This point has been added to the discussion section (L313-314 of the revised manuscript). We chose TS medium for the stress tests in order to overcome the variability of HU medium, so as to be able to modify a single parameter at a time in a controlled manner. This approach is in line with the response to the previous comment, where the aim of this study was to carry out a phenotypic screening of isolates already adapted to the urinary tract. Furthermore, as requested, the NaCl concentration of the TS medium has been indicated in the Material and Methods section of the revised manuscript (L453-454).

3/ To improve reader comprehension, it would be helpful to include a comparative schematic summarizing the main proteomic findings and highlighting the most notable differences or similarities across the six clinical isolates.

As suggested, we have added a summary diagram (Fig 10) (L294-295) grouping together the main features observed in *Pseudomonas* isolates adapted to the urinary tract, to facilitate overall understanding of the manuscript.

For instance, in Figure 7, are the differentially abundant proteins conserved across patients? What is the relative impact of TS versus HU medium on protein expression?

As rightly suggested, new Venn diagrams have been added to the supplemental material (Fig S2) which compare the protein accumulation of all late isolates and all early isolates from the three patients in urine and TS medium (L195-199). Due to the great difference in physicochemical composition between TS and HU media, and as shown by the relative distance of all proteins according to the medium in the PCA presented in the supplemental material (Fig S1), there are a few proteins conserved between patients, but it is mainly in terms of metabolic categories that many of them are found in common (L195-199).

In the same vein, including a summary table or graphic of the phenotypic test results for each of the six isolates, and comparing their progression over time, would significantly enhance the clarity of the discussion.

As requested, we have created a table summarizing the phenotypic comparisons between early and late isolates for the three patients (Table 2) (L258-259; L279-281).

Although the authors describe differentially expressed proteins under various conditions, it remains challenging to identify specific systems or pathways that should be explored in more detail. A clearer synthesis would strengthen the manuscript.

As mentioned above, we have produced a synthetic scheme (Fig 10) (L294-295) of the main characteristics related to the adaptation of *Pseudomonas* to the urinary tract.

We share the reviewer's idea to identify specific pathways that could be explored in more detail. In this context, based on our proteomic data, we highlighted that overabundance of proteins involved in flagellar assembly as well as chemotaxis-related proteins were mainly observed in early isolates and that no proteins involved in siderophore-biosynthesis were overabundant in late isolates grown in HU compared to early counterparts (L225-228 and L240-242). These observations are grouped in Table 1.

Minor Concerns:

Lines 50-53: What is the actual prevalence of *P. aeruginosa* in nosocomial UTIs? Including precise epidemiological data would be useful.

As requested, we have mentioned the actual prevalence of *P. aeruginosa* nosocomial UTI (L54-56 of the revised manuscript).

Lines 58-59: The statement that adaptation mechanisms have been "already" well studied in CF patients should be softened, as these are still under investigation.

The sentence has been edited to avoid confusion (L59-60 of the revised manuscript).

Lines 70-73 / Fig. 1: Please provide relevant patient details (e.g., age, comorbidities, site of infection, antibiotic use between isolates), which may influence evolutionary outcomes.

As mentioned in the answer to the Reviewer 1 (see comment 4), the characteristics of the patients from whom these isolates were isolated have been published previously (Cottalorda et al., 2021, Front Microbiol, DOI: 10.3389/fmicb.2020.611246). We have included in the revised version key metadata concerning patient history and antibiotic treatment history before or between isolate collection. We recalled that patients A and D were immunocompromised, that patients D and F had urinary comorbidity, and that patients A, D and F had received at least one course of antibiotics in the six months prior to collection of the first urine sample or in the period between the two samples studied (L422-425 of the revised manuscript).

Lines 93-94: Reintroduce and explain the rationale for using HU and AUM media.

We hoped to use AUM as a substitute for human urine, thus providing a stable and reproducible medium. However, various experiments showed that AUM did not adequately replicate HU (L174-176; L309-311); we therefore excluded it from further experiments.

Lines 114-116 / Fig. 3: Have the authors measured the initial osmolarity of TS? Why not use AUM, which would better represent urinary osmotic conditions?

Osmolarity of TS was not measured but the NaCl concentration was calculated to be 0.08 M. This information was included in the revised version (L453-454). As mentioned above and in the response to comment 2, we chose TS medium for stress tests in order to have reproducible data and to be able to control the biochemical parameters of the medium. Indeed, our experiments aimed to describe the phenotypes of late strains adapted to the urinary tract to possibly correlate them with changes in protein profiles.

Lines 402-403: Please clarify how AUM was prepared, including modifications to the original protocol.

As requested, the following sentence has been added to the revised version to clarify how AUM was prepared (L435-437).

Fig. 2: Generation time is reported-was final biomass (yield) also assessed?

Cell counts of the different isolates cultured for approximately 18 hours in TS and HU were carried out. The results in CFU/mL and final OD₆₀₀ are now presented in Table S1 (L94-95 of the revised manuscript).

Line 420: Was a buffering agent used to adjust pH? Was pH monitored throughout the culture period?

No buffering agent was used to adjust the pH. This point has been clarified in L455 of the revised manuscript.

Since the growths under stress conditions were carried out on microplates in small volumes, it was technically impossible to monitor the evolution of pH.

Re: Spectrum00456-25R1 (**Insights on pathoadaptation of sequential *Pseudomonas aeruginosa* isolates to the urinary tract**)

Dear Mrs. Caroline Martin-Duval:

The authors have adequately addressed all reviewer's comments.

Your manuscript has been accepted, and I am forwarding it to the ASM production staff for publication. Your paper will first be checked to make sure all elements meet the technical requirements. ASM staff will contact you if anything needs to be revised before copyediting and production can begin. Otherwise, you will be notified when your proofs are ready to be viewed.

Sincerely,
Estela Galvan
Editor
Microbiology Spectrum

Reviewer #2 (Comments for the Author):

The authors have addressed all my questions and requests.